# Structure-based prediction of T cell receptor:peptide-MHC interactions

Philip Bradley[1,2]*

[1]Herbold Computational Biology Program, Division of Public Health Sciences. Fred Hutchinson Cancer Center, Seattle, United States; [2]Institute for Protein Design. University of Washington, Seattle, United States

**Abstract** The regulatory and effector functions of T cells are initiated by the binding of their cell-surface T cell receptor (TCR) to peptides presented by major histocompatibility complex (MHC) proteins on other cells. The specificity of TCR:peptide-MHC interactions, thus, underlies nearly all adaptive immune responses. Despite intense interest, generalizable predictive models of TCR:-peptide-MHC specificity remain out of reach; two key barriers are the diversity of TCR recognition modes and the paucity of training data. Inspired by recent breakthroughs in protein structure prediction achieved by deep neural networks, we evaluated structural modeling as a potential avenue for prediction of TCR epitope specificity. We show that a specialized version of the neural network predictor AlphaFold can generate models of TCR:peptide-MHC interactions that can be used to discriminate correct from incorrect peptide epitopes with substantial accuracy. Although much work remains to be done for these predictions to have widespread practical utility, we are optimistic that deep learning-based structural modeling represents a path to generalizable prediction of TCR:pep-tide-MHC interaction specificity.

## Editor's evaluation

The study provides a significant step forward in the prediction of T cell receptor docking to peptide-major histocompatibility complex ligands using a specialised version of the deep neural network structure prediction program AlphaFold. Progress towards this goal has implications for vaccine development, and cancer immunotherapy and is an intrinsically interesting structural problem due to the variability of the T cell receptor scaffold.

*For correspondence:
pbradley@fredhutch.org

Competing interest: The author declares that no competing interests exist.

## Introduction

The specificity of T cell receptors (TCR) for peptides presented by major histocompatibility complex proteins (pMHC) is a critical determinant of adaptive immune responses to pathogens and tumors and of autoimmune disease. A predictive model of TCR:pMHC interactions, capable of mapping between TCR sequences and pMHC targets, could lead to advances in cancer immunotherapy and in the diagnosis and treatment of infectious and autoimmune diseases. Despite recent progress in TCR sequence analysis and modeling (*Gielis et al., 2019*; *Huang et al., 2020*; *Mayer-Blackwell et al., 2021*; *Montemurro et al., 2021*), a generalizable predictive model of TCR:pMHC interactions remains out of reach: existing predictors can learn to recognize new TCR sequences specific for pMHCs in their training set, but robust generalization to unseen pMHC epitopes has not been convincingly demonstrated (*Moris et al., 2021*). Two key difficulties are the diversity of TCR:pMHC recognition modes, a consequence of TCR sequence and structural diversity and flexibility in TCR:pMHC docking orientation, and the limited number of experimentally validated TCR:pMHC interaction examples for use in training.

We hypothesized that 3D structural modeling might offer a path toward generalizable prediction of TCR:pMHC interactions in the current data-limited regime. At the biophysical level, TCR:pMHC interaction specificity is determined by the structures and flexibilities of the interacting partners. A wealth of structural studies have provided valuable insights into the atomistic determinants of specificity (*Rossjohn et al., 2015*; *Rudolph et al., 2006*; *Singh et al., 2017*). Collectively, these experimentally determined structures define a range of docking geometries that likely covers the majority of unseen interactions; they also provide valuable templates for cutting-edge deep neural network structure prediction methods such as AlphaFold (*Jumper et al., 2021*) and RoseTTAfold (*Baek et al., 2021*). These prediction tools feature advanced network architectures with millions of parameters that are trained on structurally characterized proteins and their sequence homologs. Despite being trained on monomeric structures, these approaches can generate state-of-the-art structure predictions for protein complexes, and they have even been used to predict whether or not protein pairs will associate (*Humphreys et al., 2021*).

Here we show that a version of AlphaFold specialized for TCR:pMHC modeling can be used to predict TCR:pMHC binding specificity with some success. Whereas the default AlphaFold version trained to predict protein:protein docking (AlphaFold-Multimer *Evans et al., 2021*) shows inconsistent performance on TCR:pMHC structures (*Yin et al., 2022*), our specialized pipeline demonstrates improved accuracy and reduced computational cost. Moreover, this modeling pipeline has significant power to discriminate target peptides from decoy peptides as evaluated on a benchmark of human and mouse MHC class I epitopes. Importantly, success in predicting the correct peptide target correlates with structural accuracy of the models, suggesting that when the pipeline succeeds, it does so by recapitulating key specificity determinants. This work, together with previous studies applying molecular modeling techniques to TCRs (*Borrman et al., 2020*; *Jensen et al., 2019*; *Lanzarotti et al., 2018*; *Pierce and Weng, 2013*), suggests that structure-based approaches represent a promising path forward for predicting TCR:pMHC interaction specificity.

## Results
### Structure prediction

We first evaluated the structure prediction performance of a recently released version of AlphaFold (AlphaFold-Multimer *Evans et al., 2021*) that was specifically trained for protein:protein docking. AlphaFold-Multimer leverages inter-chain residue covariation observed in orthologs of the target proteins to identify amino acid pairs making interface contacts. Given that TCR:pMHC interactions are determined in part by highly variable, non-germline encoded CDR3 regions, it was unclear whether AlphaFold's strong docking performance on other systems would translate to TCR:pMHC interactions. Indeed, the AlphaFold-Multimer developers noted that it does not perform well on antibody:antigen complexes, which share many features with TCR:pMHC complexes.

We tested two versions of AlphaFold-Multimer, one in which the full sequences of the interacting partners are provided as input ('AFM_full': MHC-I or MHC-IIa, beta-2 microglobulin or MHC-IIb, peptide, TCRa, and TCRb variable and constant domains), and one in which only the directly interacting domains are provided as input ('AFM_trim': TCR constant domains, beta-2 microglobulin, and C-terminal MHC domains are removed). Restricting to the core interacting domains speeds the calculations substantially at the risk of introducing decoy docking sites at the location of interfaces with the missing domains. Although both models were capable of generating high-quality predictions on a nonredundant set of 130 TCR:pMHC complexes (as indicated by CDR loop RMSDs at and below ~2 Å; details below), prediction quality was highly variable, and visual inspection revealed that many of the predicted models had displaced peptides and/or TCR:pMHC docking modes that were outside the range observed in native proteins. Additionally, these AlphaFold predictions took multiple hours per target to complete, limiting their throughput.

One limitation of AlphaFold-Multimer is that it does not support multi-chain templates (*Evans et al., 2021*): template information from the database of solved structures can inform the internal conformation of individual chains, but it does not guide the docking of chains into higher order complexes. The constrained nature of the TCR:pMHC binding mode suggests that higher and more consistent prediction accuracy could be obtained by providing additional template information. A challenge when modeling TCR structures is that the V-alpha and V-beta genes largely determine the

best structural template, and these genes associate freely rather than in fixed pairings, which means that the optimal structural template for the TCR-alpha chain will often come from a different PDB structure as that for the TCR-beta chain. Additionally, the TCR:pMHC docking mode varies widely within an overall diagonal binding mode, in a way that is not easily predicted directly from sequence, making it challenging to select an optimal template for the TCR:pMHC relative orientation. Guided by these considerations, we developed an AlphaFold-based TCR docking pipeline that uses hybrid structural templates to provide a broad, native-like sampling of potential docking modes (*Figure 1*). In this approach, individual chain templates are first selected based on sequence similarity to the target TCR:pMHC (*Figure 1A*). Hybrid complexes are created from these individual chain templates by using a diverse set of representative docking geometries to orient the TCR chains relative to the pMHC (see Methods). Docking geometries are defined in terms of the 6 degrees of freedom that relate the MHC reference frame to the TCR reference frame, where the MHC and TCR reference frames are defined based on internal pseudo symmetry (*Figure 1B and D* and Methods). These hybrid complexes are provided as templates to multiple independent AlphaFold simulations, four templates per simulation, with the highest confidence model from the simulations taken as the final prediction (*Figure 1C*). During benchmarking, templates and docking geometries from structures with similar TCRs or pMHCs to the target are excluded to reduce bias toward the native structure (see Methods; this constraint was not applied to the default AlphaFold-Multimer methods). Given that we are providing template information that constrains the inter-chain docking, we chose not to include additional multiple sequence alignment (MSA) information beyond the target sequence. This greatly speeds the predictions: MSA building is the most time-consuming part of the AlphaFold pipeline, and the neural network inference step is also significantly faster without MSA information.

We found that the hybrid templates AlphaFold pipeline specialized for TCR:pMHC ('AF_TCR') produces higher quality models than either of the Alphafold-Multimer variants on a benchmark set (*Figure 2—figure supplement 1*) of 130 TCR:pMHC complexes (*Figure 2A*, Wilcoxon $P<10^{-7}$ vs AFM_full and $P<10^{-12}$ vs AFM_trim on the full set; *Figure 2B*, $P<10^{-3}$ for both comparisons on 20 targets without a close homolog in the AlphaFold-Multimer training set; and *Figure 2—figure supplement 2* for peptide modeling accuracy). The AF_TCR pipeline also outperforms the state of the art TCRpMHCmodels pipeline (*Jensen et al., 2019*) for Class I MHC TCR modeling (*Figure 2—figure supplement 3A–B*), and produces better docking geometries than simply borrowing the geometry from the most sequence-similar template (*Figure 2—figure supplement 3C*). There was a significant positive correlation between predicted and observed model accuracy (*Figure 2C*).

For each benchmark target, the AlphaFold TCR pipeline is provided with 12 hybrid template complexes whose TCR:pMHC docking modes are taken from 12 diverse ternary structures unrelated to the target. We were curious to know whether the AlphaFold simulation was improving on the docking information present in these template structures. To answer this question, we compared the accuracy of the docking geometry present in the final model to the accuracies of the 12 template structures. Since the 12 templates differ in the sequences and structures of their CDR loops, we developed a distance between TCR:pMHC docking geometries that compares the placement of 'generic' CDR loops ('docking RMSD', see Methods). This docking RMSD measure is correlated with CDR RMSD in comparisons of models to natives (*Figure 2—figure supplement 4*), but it focuses exclusively on the docking geometry and provides a sequence-independent way of comparing binding modes that emphasizes CDR loop placement. For 30% of the targets, the AlphaFold TCR final model had a lower RMSD than the best template docking geometry (*Figure 2D*); the final model improved over the median template RMSD for 94% of the targets (*Figure 2E*). To visualize the overall docking geometry landscape of models and natives, we calculated docking RMSD values between all of the native ternary structures and the AlphaFold-TCR and AlphaFold-Multimer models and transformed this distance matrix into a 2D projection (*Figure 2—figure supplement 5*) using the UMAP algorithm (*McInnes et al., 2018*). Inspection of this 2D docking geometry landscape reveals regions that are distant from the native structures and only sampled by the AlphaFold-Multimer models, supporting the view that incorporating template docking geometries helps to constrain predictions to native-like geometries. We analyzed the factors contributing to docking prediction accuracy and found that two dominant factors are the degree to which the docking geometry in the native structure deviates from the consensus binding mode (as captured in a multidimensional Z score, see methods) and the MHC class (class II binding modes were better predicted than class I), with minor contributions from

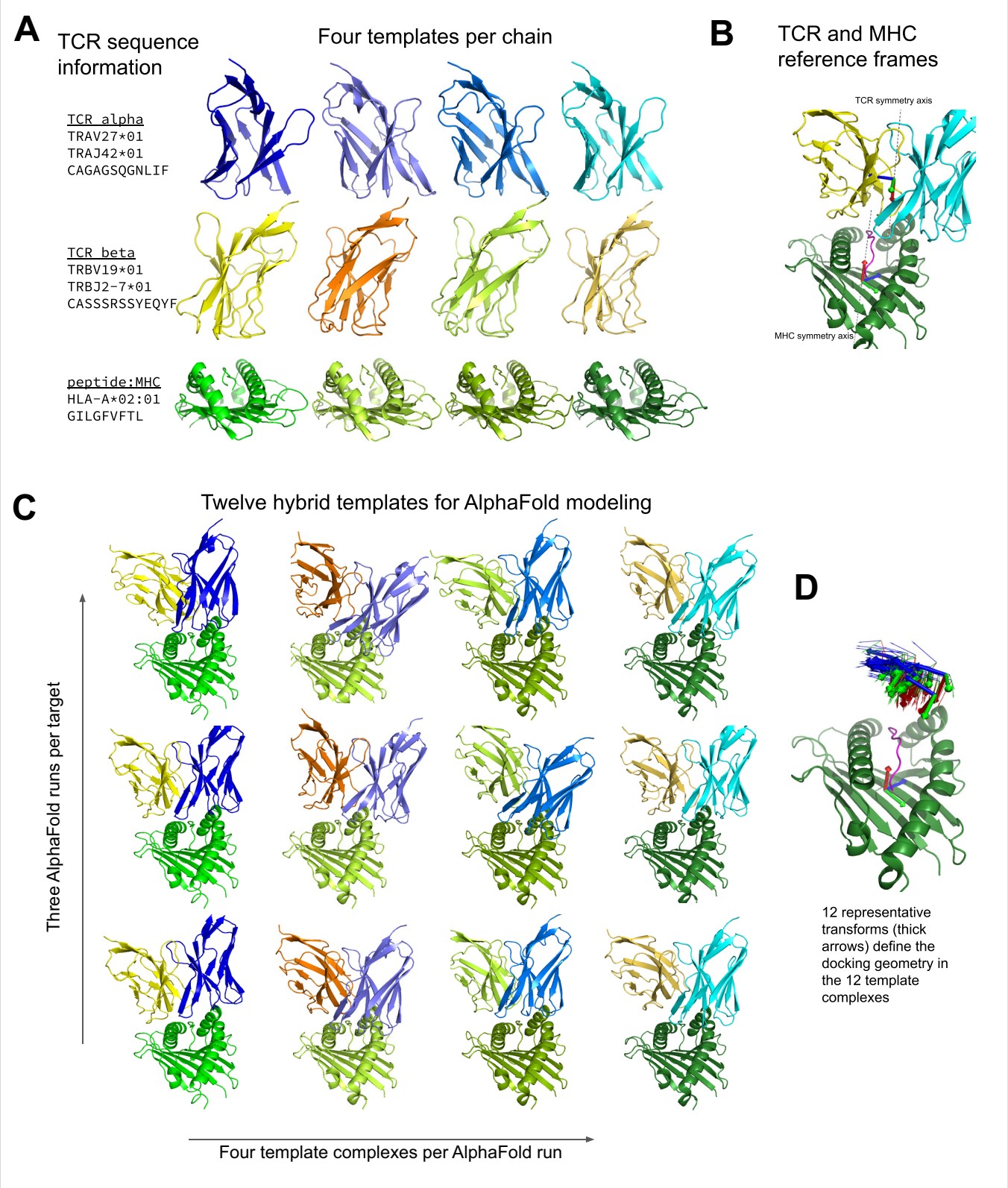

**Figure 1.** Constructing diverse hybrid templates for AlphaFold modeling. (**A**) Four structural templates for each TCR chain and for the peptide:MHC are identified in the Protein Databank (**Berman et al., 2000**) by sequence similarity search. (**B**) TCR:pMHC docking geometry is defined by computing the rigid-body transformation between TCR and pMHC coordinate frames. Coordinate frames are oriented based on internal pseudo symmetry as described in the Methods. (**C**) Three independent AlphaFold simulations are performed, each with four hybrid templates built from the four sets

*Figure 1 continued on next page*

*Figure 1 continued*

of single-chain templates oriented relative to one another using one of twelve representative docking geometries chosen to cover a wide range of experimentally determined ternary complexes. (**D**) TCR coordinate frames from class I pMHC ternary structures and the 12 representative transforms (thicker arrows) are shown in a common coordinate system defined by their corresponding pMHC coordinate frames.

The online version of this article includes the following figure supplement(s) for figure 1:

**Figure supplement 1.** MHC and TCR core residue definitions.

V gene template sequence distance, CDR loop modeling accuracy, peptide modeling accuracy, and TCRalpha/TCRbeta docking accuracy (*Figure 2—figure supplement 6*).

An attractive feature of neural network architectures is the potential to 'fine tune' a general network for improved prediction accuracy in a specific domain. We fine-tuned the AlphaFold parameters in the context of the AlphaFold TCR pipeline on the set of 93 human TCR:pMHC complexes from the benchmarking set and subsequently evaluated the performance of this model on the 37 mouse TCR:pMHC targets. Despite the small size of the TCR:pMHC ternary structure database, the fine-tuned model showed improved performance on the mouse targets (*Figure 2F*; Wilcoxon p<0.015), which are distinct in the details of their epitope, MHC, and TCR sequences from the human training set, suggesting that the model was able to learn generalizable features of TCR:pMHC interactions. This fine-tuning procedure was facilitated by the fact that the AF2 model requires significantly less memory in the absence of MSA information, making it possible to perform parameter optimization on full TCR:pMHC systems without any residue cropping.

## Binding specificity prediction

Having established that the AlphaFold TCR pipeline can generate more accurate TCR:pMHC models than AlphaFold-Multimer, we evaluated its performance in TCR epitope prediction. The general problem of predicting, de novo, which peptide:MHCs a given TCR recognizes is likely to be very difficult due to the diversity of TCR:pMHC recognition modes, the polyspecificity of individual TCRs, and the paucity of available training data (*Moris et al., 2021*). Here we consider instead the simpler problem of selecting the correct target peptide from a small set of candidates. This might correspond to a real-world scenario in which we know the source antigen from which the unknown peptide epitope is taken, or we have a positive hit in a T cell stimulation assay that implicates a pool of peptides rather than a unique epitope. For benchmarking, we focus on peptide-MHC epitopes for which a repertoire of cognate TCRs has been identified. This allows us to evaluate the sensitivity of the predictions to small changes in TCR sequence. It also lets us investigate a scenario in which we are given not one TCR, but a set of TCRs that are all predicted to recognize the same epitope, and we consider the extent to which this helps to constrain the target epitope. With improved single-cell technologies for paired TCR sequencing, and improved methods for identifying TCR sequence convergence, we hypothesize that this will become an increasingly common scenario.

We selected a set of 8 Class I peptide:MHC systems (*Table 1*) for which a repertoire of paired epitope-specific TCRs and a solved ternary structure were available. These systems include one human (A*0201) and one mouse (H2-Db) MHC allele, each with 9- and 10-residue peptides. TCR repertoires containing more than 50 unique TCR sequences were subsampled to a set of 50 TCRs using an algorithm that removed redundancy while concentrating on the more densely sampled regions of TCR space (see Methods). For each MHC/peptide length combination, we used the NetMHCpan-4.1 (*Reynisson et al., 2020*) method to select 9 decoy peptides with binding scores in the range of the true peptide binders. We additionally selected 50 irrelevant TCRs at random from human and mouse CD8 T cell datasets made available by 10 X Genomics (these TCRs were used to correct for pMHC-intrinsic effects; see below and Methods).

We used the AlphaFold TCR pipeline to generate docked complexes and associated interface accuracy estimates for pairings of each TCR with its true pMHC epitope and with 9 decoy peptides of the same length (*Figure 3A*). This produces, for each of the eight pMHCs, an Nx10 matrix of predicted interface accuracies (*Figure 3B*, left panel), where N is the number of TCRs specific for the given pMHC. To generate a single number representing the estimated interface accuracy of a complex, we summed the residue-residue predicted aligned error (PAE) for all TCR:pMHC residue pairs. These raw accuracy estimates showed significant TCR- and pMHC-intrinsic effects (*Figure 3B*). Certain TCRs had consistently higher or lower than average predicted interface accuracies due to

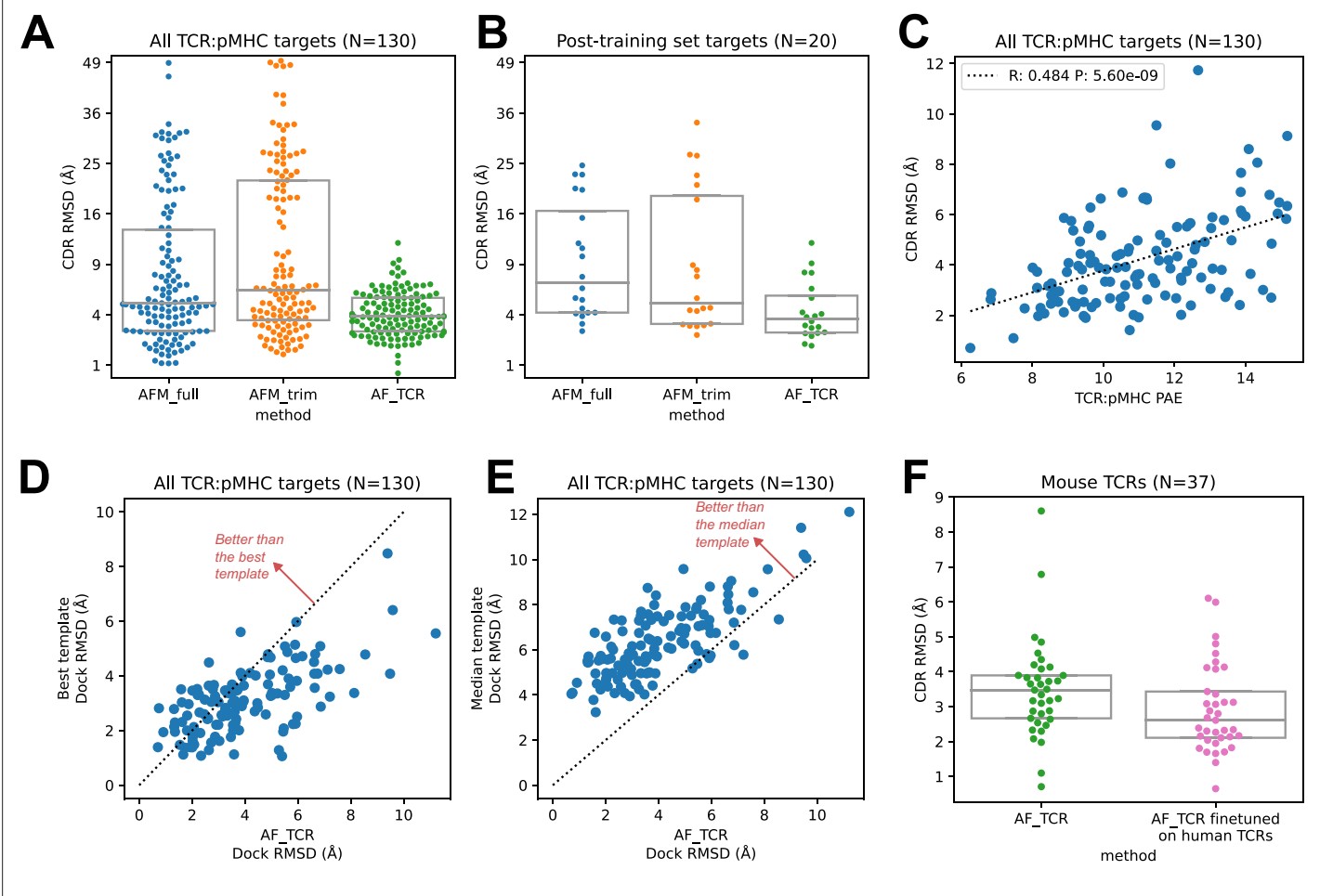

**Figure 2.** TCR modeling accuracy. (**A**) Comparison between Alphafold-Multimer with full ('AFM_full') or trimmed ('AFM_trim') input sequences and the hybrid-templates TCR pipeline ('AF_TCR'). CDR RMSD values (y-axis) are computed by superimposing the native and modeled MHC coordinates and comparing the placement of the TCR CDR loops (see Methods). (**B**) Same as in (**A**) but for the 20 benchmark targets unrelated to any TCR:pMHC structure deposited before May 2018, the cutoff date for the AlphaFold-Multimer training set. (**C**) AlphaFold's predicted aligned error (PAE) measure, evaluated between TCR and pMHC, correlates with CDR RMSD between model and native structure. (**D**) The docking geometry of the final AlphaFold model improves over the best of the 12 templates in 30% of cases (points above the line *y=x*). (**E**) The docking geometry of the final AlphaFold model improves over the median of the 12 templates in 94% of cases (points above the line *y=x*). (**F**) Fine-tuning AlphaFold's parameters on human TCR:pMHC complexes improves prediction of mouse TCR:pMHC complexes. Boxes in A, B, and F show the quartiles of the plotted distributions.

The online version of this article includes the following source data and figure supplement(s) for figure 2:

**Source data 1.** Structure prediction benchmark.

**Figure supplement 1.** Composition of the TCR:pMHC ternary structure database.

**Figure supplement 2.** Peptide structural modeling accuracy.

**Figure supplement 3.** TCR:pMHC modeling performance.

**Figure supplement 4.** Comparison of docking RMSD to CDR RMSD.

**Figure supplement 5.** Docking geometry landscapes for the structure prediction benchmark.

**Figure supplement 6.** Factors influencing AF_TCR docking accuracy.

features such as longer CDR3 loops or usage of V genes without a close structural template. We saw similar, albeit weaker, trends across different peptide:MHC complexes, perhaps due to AlphaFold's confidence in the MHC-bound structure of the peptide. TCR-intrinsic factors do not change the relative order of candidate peptides, but they make comparisons of binding predictions across TCRs difficult; pMHC effects have the potential to change the rank ordering of candidate peptide epitopes. Since we are interested here in evaluating the compatibility between TCR and pMHC and not, e.g.,

**Table 1.** Binding specificity benchmark.

| Organism | MHC | Peptide length | Peptide sequence | Antigen |
|---|---|---|---|---|
| human | HLA-A*02:01 | 9 | GILGFVFTL | Flu M1 |
| human | HLA-A*02:01 | 9 | GLCTLVAML | EBV BMLF1 |
| human | HLA-A*02:01 | 9 | NLVPMVATV | CMV pp65 |
| human | HLA-A*02:01 | 9 | YLQPRTFLL | SARS-CoV-2 Spike |
| human | HLA-A*02:01 | 10 | ELAGIGILTV | human MART-1 |
| human | HLA-A*02:01 | 10 | KLVALGINAV | HCV POLG |
| mouse | H2-Db | 9 | ASNENMETM | Flu NP |
| mouse | H2-Db | 10 | SSLENFRAYV | Flu PA |

ranking peptides by their affinity for MHC, we corrected for these TCR- and pMHC-intrinsic effects to generate an array of TCR:pMHC binding scores intended to be comparable across different pMHCs and TCRs (*Figure 3B*, middle panel; lower scores indicate stronger predicted binding, see Methods).

We evaluated the accuracy of these binding predictions across the eight pMHC epitopes. First, we calculated the rank of the true peptide epitope amongst the 9 decoy peptides (*Figure 3B*, right panel) on a per-TCR basis. To visualize how these ranks vary across each pMHC-specific repertoire, we constructed hierarchical clustering trees of the TCR sequences using the TCRdist measure (*Dash et al., 2017*) and colored them by the rank of the true peptide (*Figure 3C* and *Figure 4*). Internal edges, which correspond to multiple 'leaf' TCRs, are colored by the rank of the true peptide after averaging the binding scores over the leaf TCRs. Looking across all eight epitopes, we can see, first, that the predictions are not random: on average the correct peptide is ranked more favorably than most of the decoys (i.e. there is more blue than red). For six of the eight epitopes, the correct peptide is ranked first when we average the binding scores of all the TCRs in the repertoire (*Figure 3D*; *Figure 4*: the largest branch of the tree is dark blue). It also appears that the epitopes with more sequence-diverse repertoires (A*0201-GLC9 and A*02:01-NLV9) are more challenging to predict: the trees that merge completely at smaller TCRdist values (further to the left) are bluer than the other trees in *Figure 4*. This can be seen quantitatively by plotting the TCRdiv repertoire sequence diversity measure (*Dash et al., 2017*) against measures of binding prediction success (*Figure 4—figure supplement 1*). If we rank the peptides by binding score and compare the recovery of true binder peptides to decoys using receiver operating characteristic (ROC) curves, we can see that some epitopes, such as A*02:01-YLQ9 and A*02:01-ELA10 are predicted very well (by area under the ROC curve, AUROC ≥ 0.96) and some predictions are only slightly better than random (*Figure 3E*). We find an overall AUROC value of 0.82 when binding and non-binding TCR:pMHC pairs from all epitopes are ranked together.

We looked to see whether structural modeling accuracy correlated with binding prediction success (*Figure 5*). Although very few of the specific TCRs being modeled have been structurally characterized, each of the epitopes has at least one solved ternary structure in the protein structure database. For each TCR, we computed docking RMSDs between the TCR:pMHC model in complex with its cognate epitope and the solved ternary structures for that epitope and took the minimum value as a proxy for the accuracy of the predicted binding mode. *Figure 5A* shows the distribution of these RMSD values across each repertoire. Well-predicted epitopes such as A*02:01-YLQ9 and A*02:01-ELA10 indeed appear to have smaller RMSD values than other repertoires. The mouse pMHC H2Db-ASN9 is an outlier, with an RMSD distribution shifted to very high values. Examination of the three ternary structures for this pMHC revealed that they represent a unique population of TRBV17+ TCRs that is distinct from the consensus repertoire modeled here. Two of the three TCRs bind with a reversed docking orientation (*Gras et al., 2016*), and the third has a highly displaced binding footprint (*Zareie et al., 2021*); all three are outliers in a hierarchical clustering tree of Class I TCRs based on docking RMSD (*Figure 5—figure supplement 1*). If we exclude H2Db:ASN9 and plot docking RMSD to the closest epitope structure versus binding score for the correct peptide, we see that there is a positive correlation (*Figure 5B*). The TCRs for which the correct peptide is ranked first have a lower RMSD distribution than other TCRs, and this RMSD distribution shifts upward as the rank of the correct peptide declines (*Figure 5C*). These results suggest that the correct binding predictions are driven

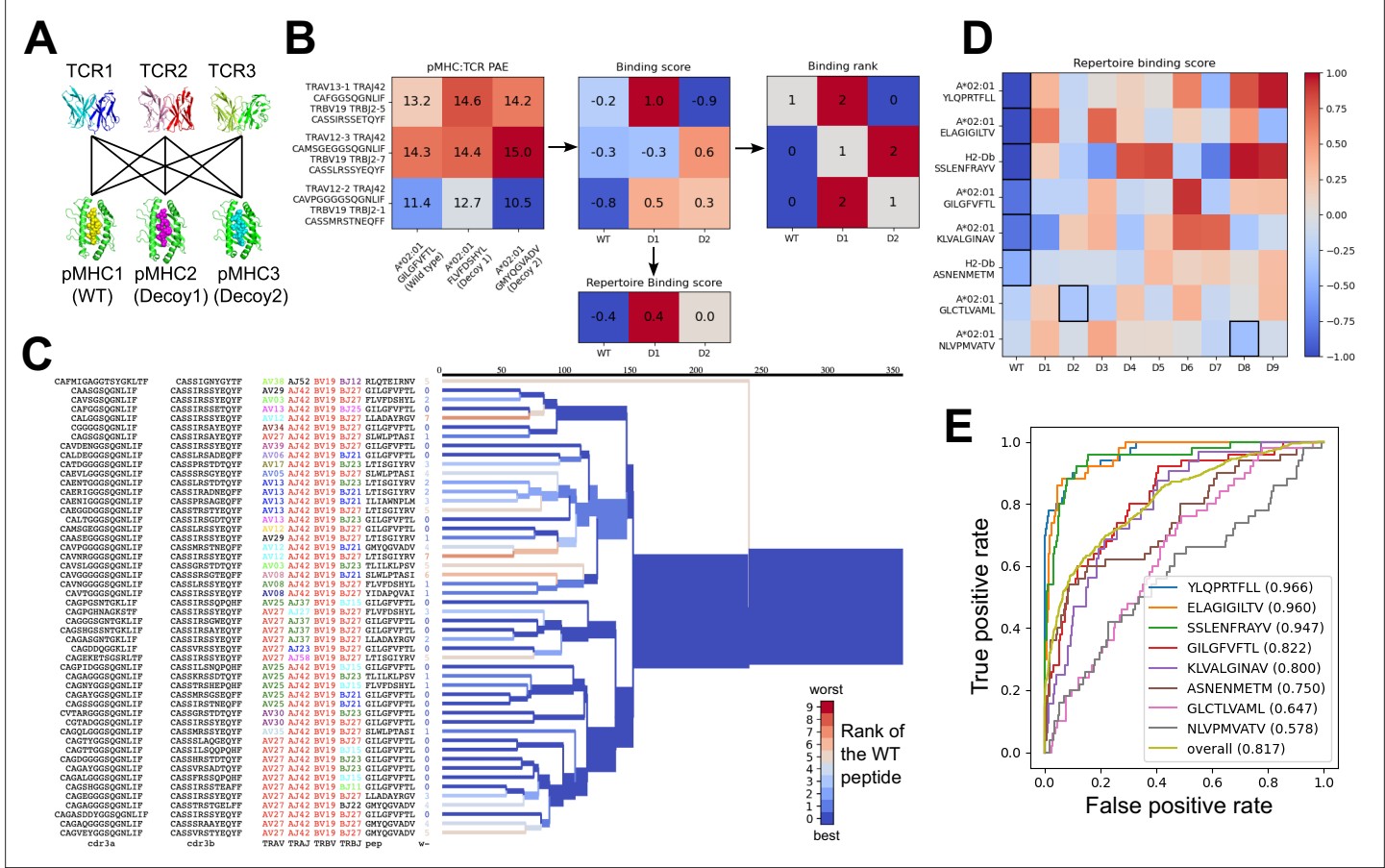

**Figure 3.** Structural modeling can sometimes discriminate correct from incorrect TCR:pMHC pairings. (**A**) For each of the eight peptide:MHC epitopes, we docked multiple cognate TCRs against multiple decoy peptides and the wild type epitope. Here three TCRs and three pMHCs are shown; 9 decoys and up to 50 TCRs were actually modeled. (**B**) For each candidate TCR:pMHC pairing, the mean AlphaFold predicted aligned error (PAE) for the TCR:pMHC interface was calculated (left) and transformed into a binding score by subtracting out TCR-intrinsic and pMHC-intrinsic factors (middle). These binding scores were averaged to define a repertoire-level binding score for the WT epitope and each of the decoys (bottom). Also calculated was the rank of the WT binding score within the list of all the binding scores for each TCR (right). (**C**) TCRdist hierarchical clustering tree of the 50 modeled TCRs for the A*02:01 GIL9 epitope, labeled with the TCR sequence information, top-ranked peptide, and rank of the WT peptide, and colored by the rank of the WT peptide. Internal edges, which correspond to multiple 'leaf' TCRs, are colored by the rank of the WT peptide after averaging the binding scores over the leaf TCRs. (**D**) Repertoire binding scores for each of the eight target epitopes and the 9 decoy peptides, with the lowest (most favorable) binding score in each row boxed. (**E**) Receiver operating characteristic (ROC) curves for discrimination of WT from decoy peptides by binding score. Area under the ROC curve (AUROC) values are given in the legend along with the sequence of the WT peptide.

The online version of this article includes the following source data for figure 3:

**Source data 1.** Epitope specificity benchmark TCRs.

**Source data 2.** Epitope specificity benchmark peptides.

at least in part by recovery of native-like structural features (analysis of peptide backbone RMSDs shows a positive, but much weaker, correlation between binding prediction and modeling accuracy: *Figure 5—figure supplement 2*).

To further investigate the behavior of our modeling approach, we performed an in silico epitope alanine scan of each of the eight pMHC-specific repertoires. We built models and calculated binding scores for each epitope-specific TCR docked to all single-alanine mutants of the native peptide (native alanine residues were mutated to glycine). Binding scores for each TCR and each of the alanine mutants are shown in the heatmaps in *Figure 6*. Averaging these binding scores over all of the TCRs for each epitope and subtracting the score for the native peptide gives a predicted repertoire-level sensitivity to mutation at each peptide position (*Figure 6B*). From these sensitivity plots, we can see that the majority of the epitope-specific repertoires show the expected preference for the native

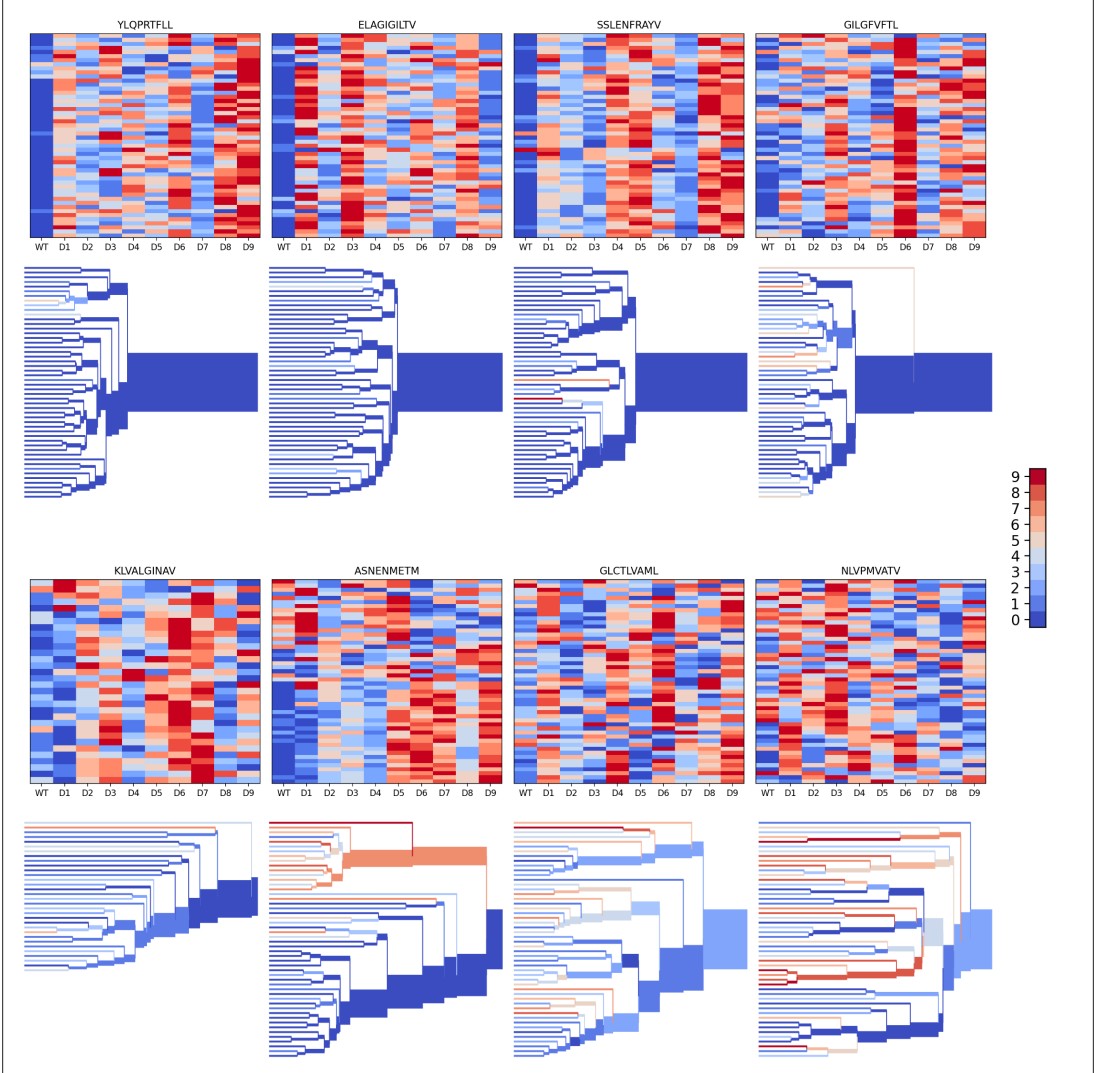

**Figure 4.** Peptide decoy discrimination results for the eight benchmark epitopes. The rank of the wild type peptide relative to the 9 decoys (0=best, 9=worst) is shown in a heatmap and a TCRdist hierarchical clustering tree of the epitope-specific TCRs. Each row of the heatmap corresponds to a single TCR; each column corresponds to one of the 10 modeled peptides, with the wild type peptide on the left. The vertical ordering of the TCRs in the heatmaps and trees is the same. Internal edges of the trees, which correspond to multiple 'leaf' TCRs, are colored by the rank of the wild type peptide after averaging the binding scores over the leaf TCRs.

The online version of this article includes the following figure supplement(s) for figure 4:

**Figure supplement 1.** Peptide specificity prediction accuracy is inversely correlated with repertoire sequence diversity.

peptide at most positions, with a subset of positions showing high sensitivity. Coloring the pMHC structures by mutation sensitivity (*Figure 6A*) reveals that these highly sensitive positions are largely TCR-exposed; several are sites of known viral escape mutations, such as A*02:01-KLV position L5 (*Wölfl et al., 2008*) and H2Db-SSL position R7 (*Valkenburg et al., 2013*). Although the observation that positions predicted to disrupt TCR binding are largely TCR-exposed accords with biophysical intuition, this is still an important validation of the protocol. Since the binding scores are derived from pairwise AlphaFold confidence measures partly involving the peptide, one concern is that they might be reflecting peptide-MHC binding preferences rather pMHC-TCR binding. The fact that peptide anchor mutations are not among the most strongly predicted positions here suggests that, by subtracting each peptide's average binding score for the background 'non-binder' TCRs, we are able to correct for these peptide-intrinsic features. As a final test, we evaluated the specificity protocol in a more challenging setting: single TCRs (rather than TCR repertoires) interacting with altered peptide

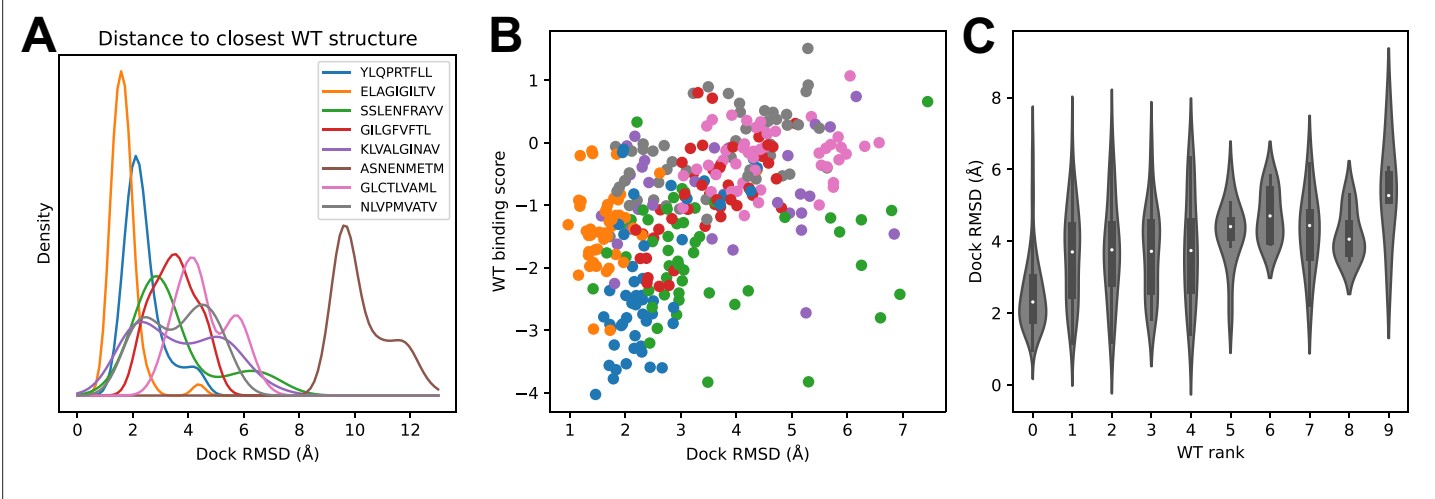

**Figure 5.** Success in decoy discrimination correlates with structural modeling accuracy. (**A**) For each TCR, the structural model in complex with the wild type epitope was compared to all experimentally determined ternary structures for that epitope and the smallest docking RMSD was recorded. The resulting RMSD distributions were smoothed using kernel density estimation and plotted. (**B**) Scatter plot of docking RMSD to the nearest wild type structure versus the binding score for the wild type peptide. Favorable wild type binding scores correlate with lower RMSD values. (**C**) Distributions of docking RMSD to the nearest wild type structure (y-axis) as a function of the rank of the wild type peptide (x-axis). When the wild type peptide is ranked first (left violin), the corresponding docking geometries are more similar to those of ternary complexes for that epitope, suggesting higher accuracy.

The online version of this article includes the following figure supplement(s) for figure 5:

**Figure supplement 1.** Hierarchical clustering tree of TCR:pMHC class I docking geometries.

**Figure supplement 2.** Peptide backbone accuracy in the specificity benchmark.

variants, some with diverse amino acid mutations. Here we found much poorer performance than in the decoy-discrimination task (*Figure 6—figure supplement 1*), which could be somewhat improved by including related TCR:pMHC complexes in the template pool (suggesting that future improvements to the structure prediction methods may translate into improved binding predictions).

## Discussion

Prediction of TCR:pMHC interactions is challenging because of the diversity of TCR:pMHC recognition modes and the limited number of validated interactions available for training. Inspired by recent breakthroughs in protein structure prediction (*Baek et al., 2021*; *Jumper et al., 2021*), we hypothesized that structure-based approaches, which can leverage general features of protein structures and interactions, might offer a path to generalizable TCR:pMHC binding predictions from limited data. We developed a specialized AlphaFold pipeline for TCR:pMHC structure prediction that uses hybrid templates assembled from existing TCR:pMHC structures to constrain the TCR docking orientation to native-like geometries. Here we show that this pipeline can generate more accurate structure predictions of TCR:pMHC complexes than the state-of-the-art method Alphafold-Multimer. Prediction accuracy correlates with model confidence, and model quality can be further improved by fine-tuning the AlphaFold parameters on TCR:pMHC structures. When tested on peptide decoy discrimination, we found that the model's docking accuracy estimates, corrected for TCR- and pMHC-intrinsic effects, could be used to select the correct target peptides from decoys with substantial accuracy. Success in this decoy discrimination task correlated with the structural accuracy of the models, suggesting that the pipeline was picking out the correct peptide on the basis of molecular specificity determinants. Prediction accuracy varied across pMHC epitopes, with those epitopes having more sequence-diverse TCR repertoires proving more challenging to model.

There are a number of caveats to this work. First, the overall level of accuracy falls short of what would be required for most practical applications of TCR:pMHC binding prediction. As described below, we are pursuing multiple avenues for improving this initial pipeline; it may also be possible to predict from the simulations themselves which systems are reliably modeled, which could allow

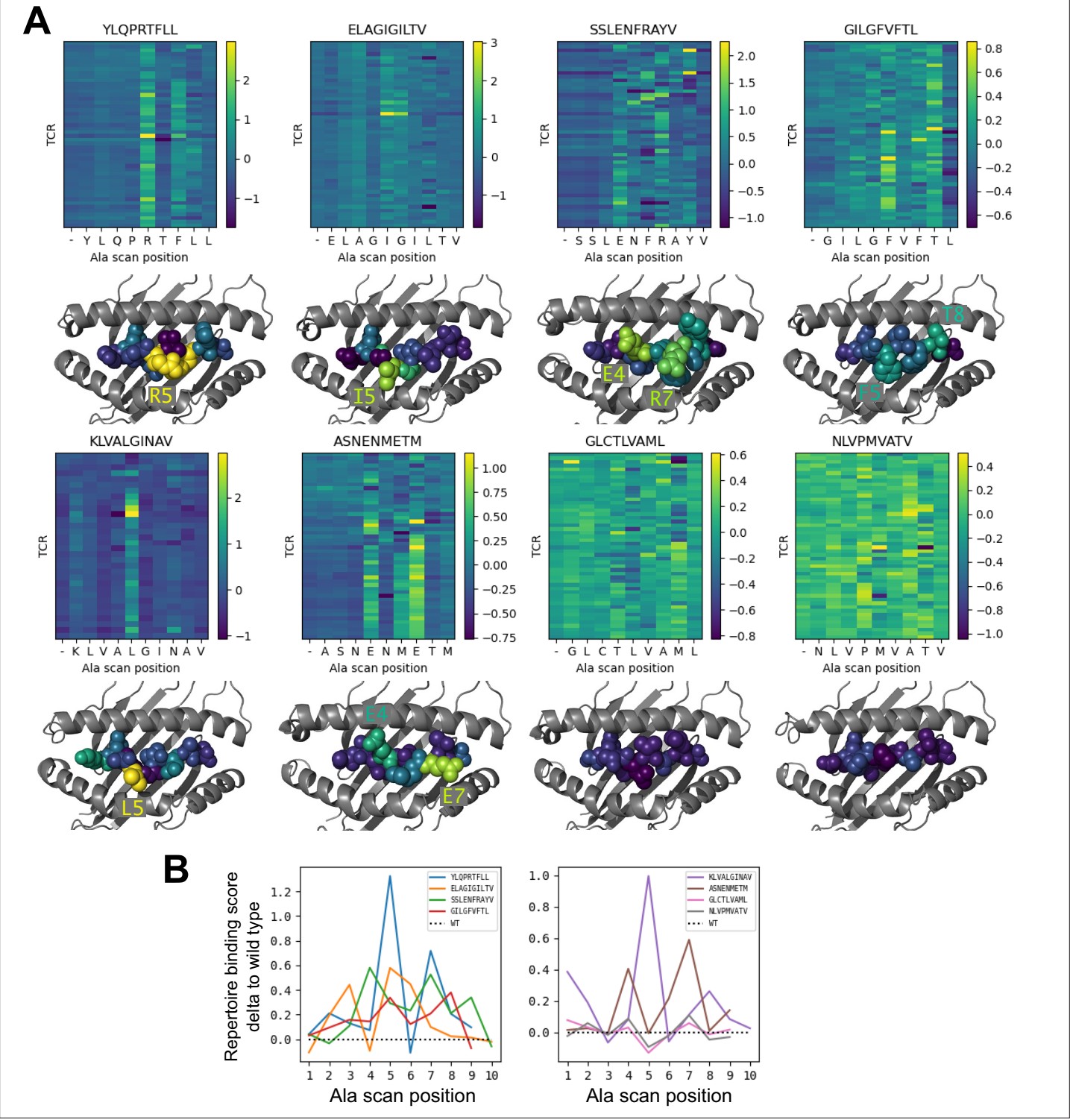

**Figure 6.** Alanine scanning results for the eight benchmark epitopes. (**A**) Heatmaps showing the binding scores for the wild type peptide (left column) and all single-alanine mutants (columns labeled with the wild type sequence) in complex with each TCR (rows). Below each heatmap, the wild type pMHC crystal structure is shown with the peptide colored by the delta between mutant and wild-type repertoire-averaged binding scores. (**B**) Line plots of the delta between the mutant and wild-type repertoire-averaged binding scores reflect the predicted repertoire-level sensitivity to epitope mutations.

The online version of this article includes the following figure supplement(s) for figure 6:

**Figure supplement 1.** Comparison to experimental data on single TCRs binding altered peptide ligands.

useful predictions to be extracted from large-scale calculations. Second, several of the epitopes in our peptide decoy discrimination benchmark have been extensively characterized in structural studies. While we made efforts to avoid using information from related structures during template assembly (see Methods), it is still possible that bias toward native-like conformations was introduced. For example, the AlphaFold parameters that we rely on in the pipeline were trained on individual protein chains (not protein complexes) deposited prior to May 2018. Some of the TCR chains modeled in the decoy discrimination task are likely similar to protein chains present in this AlphaFold training set. As the database of TCR:pMHC pairs grows, future benchmarking will establish whether the performance observed here will extend to epitopes without structural coverage. Until then, these results should be considered a favorable upper bound on the method's performance. Finally, our template-based modeling approach is unlikely to succeed on TCR:pMHC systems with highly divergent binding modes. Although we do see evidence that AlphaFold can improve over the best template provided (*Figure 2D*), it is unlikely that it can reliably predict complexes that deviate substantially from any template (e.g. reversed-orientation geometries *Beringer et al., 2015*; *Gras et al., 2016*). More generally, a template-based approach such as ours is fundamentally limited by the coverage of the structure database, which is highly biased toward well-studied alleles such as HLA-A*02:01 and, for MHC Class I, toward canonical 9-residue epitopes (*Figure 2—figure supplement 1*).

The modeling pipeline described here represents a first step in applying deep learning structure prediction tools to study TCR:pMHC interactions. We anticipate that it can be improved by further testing on other systems and by comparison with other experimental data types (binding affinities, interface mutations, etc.). This initial pipeline does not make use of multiple sequence alignment (MSA) information, but it may be helpful to include MSAs for individual chains or to construct 'paired MSAs' consisting of concatenated TCR:peptide:MHC sequences of known binding examples. Such paired MSAs could take the place of the paired ortholog alignments used by AlphaFold-Multimer to detect residue covariation across interfaces. We evaluated the use of AlphaFold's residue-residue accuracy estimate (PAE) to discriminate wild type from decoy peptide-MHC epitopes, but it may also be worth exploring the use of other binding affinity estimates such as binding energies computed with the Rosetta software package (*Leaver-Fay et al., 2011*) or other molecular modeling tools (*Lee et al., 2018*). Finally, it may be possible to fine-tune AlphaFold parameters directly to discriminate TCR:pMHC binding examples from non-binding examples, as we have recently demonstrated for peptide:MHC interactions (*Motmaen et al., 2022*). This would allow us to directly leverage the thousands of validated TCR:pMHC interactions within the context of a structurally informed training procedure.

## Methods
### Defining TCR:pMHC docking geometry

The TCR:pMHC docking geometry is defined by the rigid body transformation that maps between the MHC and TCR coordinate frames (*Figure 1B*). The MHC coordinate frame is defined on the basis of the approximate 2-fold symmetry axis that relates the N- and C-terminal halves of the beta sheet forming the floor of the peptide binding pocket. 12 core residues in the beta sheet were selected (*Figure 1—figure supplement 1A*), 6 from the N-terminal half and 6 from the C-terminal half, that are related by this approximate 2-fold rotational symmetry. For a given MHC structure, the transformation mapping these 12 residues onto themselves, interchanging the N- and C-terminal residues and minimizing the RMSD of the alpha carbon atoms, is computed. The rotation axis of this orthogonal transformation, oriented to point toward the peptide, is taken as the x-axis of the MHC coordinate frame. The z-axis of the coordinate frame points from the center of mass (COM) of the 6 N-terminal core alpha carbons to the COM of the 6 C-terminal core alpha carbons. The coordinate frame is centered at the COM of the 12 core residues.

To define the TCR coordinate frame, 13 structurally conserved core residues from the TCR alpha chain and 13 aligned core residues from the TCR beta chain (*Figure 1—figure supplement 1B–C*) were selected on the basis of visual inspection of TCR multiple structural alignments. The same procedure as outlined above for the MHC is used to define the TCR coordinate frame, replacing the 6 N-terminal and 6 C-terminal core residues of the MHC with the 13 TCRA and 13 TCRB core residues of the TCR heterodimer. The x-axis of the coordinate frame is chosen to point along the TCR pseudo

symmetry axis toward the CDR loops, while the z-axis points from the COM of the TCRA core residues to the COM of the TCRB core residues.

The docking geometry is defined by the rigid body transformation relating the MHC coordinate frame to the TCR coordinate frame. This transformation naturally lives in a 6-dimensional non-Euclidean space (SE(3)). We take advantage of the fact that, as defined above, the x-axes of the MHC and TCR frames point toward the typical location of their partner in order to define a local 6-dimensional parameterization of this space in terms of the distance between the frame origins, a dihedral angle about the axis connecting the frame origins, the unit vector pointing from the MHC to TCR in the MHC frame, and the unit vector pointing from TCR to MHC in the TCR frame (see the README at https://github.com/phbradley/TCRdock for further details and visualizations). This mapping of TCR:pMHC docking geometries to 6 real-valued parameters allows us to approximate the space of docking geometries by a multidimensional normal distribution and assign a 'Z score' (using the Mahalanobis distance) to any observed docking geometry. This score reflects the degree to which the docking geometry diverges from the consensus binding mode for its MHC class and was found to be a strong predictor of docking accuracy (*Figure 2—figure supplement 6* upper left panel). The Python script parse_tcr_pmhc_pdbfile.py in the TCRdock github repository (see Code Availability) computes the MHC and TCR coordinate frames for an input PDB structure and calculates the docking geometry.

## AlphaFold modeling pipeline

To model a given TCR:pMHC target, three AlphaFold simulations (using the 'model_2_ptm' parameter set) are conducted and the final model with the lowest predicted aligned error (PAE) between the TCR and pMHC is selected (*Figure 1*). The model_2_ptm parameter set was chosen based on our experience in peptide:MHC binding predictions, but the model_1_ptm set gives very similar results. To reduce parameter training bias, we used the original AlphaFold monomer parameters, which were trained on single protein chains, rather than the AlphaFold-Multimer parameter set, whose training set included protein complexes. Each AlphaFold simulation can use a maximum of four templates, allowing for 12 total templates across the three runs (*Figure 1C*). These 12 templates are constructed from four templates for each of the pMHC, TCRA, and TCRB chains selected on the basis of sequence identity to the modeling target (*Figure 1A*) combined with 12 docking geometry templates. The same four templates per chain are used in each of the three AlphaFold runs; only the docking geometries vary between runs. Thus the full combinatorics of chain templates by docking geometries is not sampled. Peptide-MHC templates are sorted by total sequence identity computed over both the MHC and the peptide. To create hybrid templates for AlphaFold modeling, the pMHC and TCRB template coordinates must be mapped into the coordinate frame of the TCRA template structure. First, the TCR structure from which the TCRB template coordinates are being taken is superimposed onto the TCRA template structure by superimposing the 13 TCRA core residues. Then the superimposed TCRB coordinates are appended to the hybrid template after the TCRA coordinates. To map the pMHC coordinates into the coordinate frame of the TCRA and TCRB coordinates, MHC and TCR coordinate frames are defined as described above, and 12 representative docking geometries are selected. Each docking geometry defines the transformation between the MHC and TCR coordinate frames, allowing the pMHC template coordinates to be mapped into the hybrid template TCR coordinate frame. To choose the 12 representative docking geometries, docking geometries from TCR:pMHC structures of the same MHC class as the target are hierarchically clustered and the clustering tree is cut at a distance threshold at which there are 12 clusters. The docking geometry from each cluster with the smallest mean distance to the other cluster members is chosen as the representative. For hierarchical clustering, a matrix of docking RMSDs (defined below) is provided to the hierarchy.linkage function in the SciPy (*Virtanen et al., 2020*) cluster module. The hierarchy.fcluster function with 'maxclust' criterion is used to select the distance threshold at which the docking geometry tree divides into 12 clusters. Template structures were downloaded from the RCSB Protein Databank (*Berman et al., 2000*) ftp site on 2021-08-05.

## Fine-tuning AlphaFold for TCR:pMHC structure prediction

To fine tune the AlphaFold parameters for TCR:pMHC structure prediction, we used a version of the AlphaFold package that was modified slightly to expose the parameter training interface (*Motmaen*

*et al., 2022*). The Python script run_finetuning_for_structure.py in the alphafold_finetune github repository (https://github.com/phbradley/alphafold_finetune; *Bradley, 2022a*) with the additional command line flags '--model_name model_2_ptm --crop_size 419' was provided with a training set consisting of three runs for each of the 93 human ternary structures (279 total training examples). Due to the small size of the training dataset, training was stopped after two epochs to avoid over-fitting.

## Structure prediction benchmark

The structure prediction benchmark set consists of 130 nonredundant ternary TCR:pMHC structures deposited prior to 2021-08-05 (*Figure 2—source data 1*). No two structures in the set have fewer than 3 peptide mismatches *and* a paired TCRdist (*Dash et al., 2017*) distance less than or equal to 120. This constraint eliminates pairs of structures with the same or similar TCRs binding to the same or similar peptides. After visual inspection, we eliminated the following 9 outlier structures with highly divergent binding modes (reversed docking orientations, extremely bulged peptides, etc.): PDB IDs 5sws, 7jwi, 4jry, 4nhu, 3tjh, 4y19, 4y1a, 1ymm, and 2wbj.

During benchmarking, we excluded templates and docking geometries that were too similar to the target sequence being modeled. Peptide-MHC templates were excluded if they had fewer than three peptide mismatches with the target peptide. TCR chain templates were excluded if they had a single-chain TCRdist of 36 or less to the target chain (corresponding to three non-conservative mismatches or indels in the CDR3 loop). Docking geometries were excluded if they came from a structure with fewer than three peptide mismatches to the target or a TCRdist of 48 or less from the target TCR.

## RMSD measures

We assessed model accuracy by comparing the placement of the CDR loops relative to the MHC in the native and modeled structures. The two structures were first superimposed on the MHC coordinates; then an alpha-carbon RMSD was calculated (without further superposition) over the CDR loops, up-weighting residues in the CDR3 by a factor of 3 to reflect the greater importance of the CDR3 for epitope recognition (this is the 'CDR RMSD' reported in *Figure 2*). TCRdist CDR loop definitions were used.

To compare docking geometries between structures with different CDR loop sequences, we developed a 'docking geometry RMSD' intended to approximate the CDR RMSD in a sequence-independent fashion. The full template database was first used to calculate a mean center of mass of the residues in each CDR loop with respect to the TCR coordinate frame. To compute the docking RMSD between two docking geometries, each docking geometry is used to build a TCR coordinate frame assuming the MHC coordinate frame is centered at the origin and aligned with the coordinate axes. Then the CDR centers of mass are built with respect to each of these two TCR coordinate frames, and an RMSD is calculated between these two sets of eight points (4 CDR centers of mass each for the TCRA and TCRB chains) without superposition, upweighting the CDR3 center of mass by a factor of 3. The correlation between CDR RMSD and docking RMSD is shown in *Figure 2—figure supplement 4*.

## Epitope decoy discrimination benchmark

Eight MHC class I epitopes with TCR repertoire data and experimentally determined structures were selected as targets for a decoy discrimination benchmark (*Table 1*). Paired alpha and beta sequences of TCRs specific for these eight epitopes were collected from the literature (*10xGenomics, 2020*; *Dash et al., 2017*; *Francis et al., 2022*; *Minervina et al., 2022*; *Schattgen et al., 2022*; *Shugay et al., 2018*). Epitope-specific TCR repertoires with more than 50 TCRs were subsampled to 50 representatives using a Gaussian kernel density-based algorithm designed to preferentially sample denser regions of TCR space without introducing excessive redundancy (see algorithms_from_the_paper.py in the TCRdock github repository). The goal in sampling denser regions of TCR space was to avoid outlier TCR sequences that might represent experimental errors. 100 additional 'irrelevant' background TCR sequences (50 mouse TCRs and 50 human TCRs) were selected at random from naive CD8 T cells in datasets made publicly available by *10xGenomics, 2020* for human and here for mouse. All epitope-specific and background TCR sequences are listed in *Figure 3—source data 1*.

The eight MHC class I epitopes include 9 and 10 residue peptides presented by the MHC alleles HLA-A*02:01 and H2-Db. For each MHC and peptide length, 9 decoy peptides were selected by

scanning a 1500 residue artificial source antigen sequence with NetMHCpan-4.1 (*Reynisson et al., 2020*) and selecting the top 9 predicted binders (*Figure 3—source data 2*). The artificial source antigen sequence was created by concatenating the source antigen sequences for the nine benchmark targets (*Table 1*), shuffling, and selecting the first 1500 residues.

Each epitope-specific TCR was modeled in complex with its cognate peptide epitope and in complex with the nine length- and MHC-matched decoy peptides using the AlphaFold pipeline specialized for TCRs. The mean predicted aligned error (PAE) residue-residue accuracy measure for TCR:pMHC residue pairs was calculated for each complex and stored in an Nx10 matrix, where N is the number of TCRs (each row corresponds to a TCR and each column to a peptide). To convert these raw TCR:pMHC PAE values into a binding score that can be compared across TCRs and pMHCs, we also modeled each pMHC in complex with 50 irrelevant background TCRs from the same organism. The mean TCR:pMHC PAE for these background complexes was calculated for each pMHC and was subtracted from the matrix column of PAE values involving that pMHC. The values in the resulting matrix of adjusted PAE values were then shifted to have 0 row sums by subtracting its mean value from each row. Thus in the final Nx10 matrix of binding scores, the mean value for each row is 0, while the mean values of the columns reflect the overall binding preference of the full repertoire of TCRs for the peptide corresponding to the column ('Repertoire binding score' in *Figures 3 and 6*).

During modeling, the TCR- and pMHC-similarity constraints described above in 'Structure Prediction Benchmark' were applied to exclude templates; in addition, ternary structures with a peptide having fewer than three mismatches from the wild type peptide were excluded from all simulations (with decoy or wild type peptides). Note that the original AlphaFold monomer network (model_2_ptm), not the structure fine-tuned network, was used for the epitope specificity benchmark, since the training set used for fine-tuning overlapped with the specificity benchmark targets.

The epitope alanine-scanning benchmark was performed as described above with the difference that the decoys were single-residue alanine mutants of the wild type peptide (alanine residues in the wild type peptide were mutated to glycine). Thus there were nine decoys for 9-residue peptides and 10 decoys for 10-residue peptides.

## Software and data availability

Python software to set up and run the TCR-specialized AlphaFold pipeline described here and to parse TCR:pMHC ternary structures are available in the TCRdock github repository (https://github.com/phbradley/TCRdock, copy archived at swh:1:rev:060bdb4a59391f2d7d57b0f2a923e4b4d6c9a89f; *Bradley, 2022b*). Benchmark datasets are provided as Source Data for *Figures 2 and 3*.

## Acknowledgements

I am grateful to Jeremy Crawford, Anastasia Minervina, Amir Motmaen, Paul Thomas, and Albert Yeh for helpful comments on the manuscript, to Justas Dauparas for help fine-tuning AlphaFold, to the creators of AlphaFold for freely sharing their software and parameters, and to Fred Hutch Scientific Computing and NIH ORIP S10OD028685 for outstanding computing infrastructure. This research was supported by NIH grants R35 GM141457 and R01 AI136514.

## Additional information

### Funding

| Funder | Grant reference number | Author |
| --- | --- | --- |
| National Institutes of Health | R35 GM141457 | Philip Bradley |
| National Institutes of Health | R01 AI136514 | Philip Bradley |

The funders had no role in study design, data collection and interpretation, or the decision to submit the work for publication.

## Author contributions
Philip Bradley, Conceptualization, Resources, Data curation, Software, Formal analysis, Supervision, Funding acquisition, Validation, Investigation, Visualization, Methodology, Writing - original draft, Project administration, Writing - review and editing

## Author ORCIDs
Philip Bradley http://orcid.org/0000-0002-0224-6464

## Decision letter and Author response
Decision letter https://doi.org/10.7554/eLife.82813.sa1
Author response https://doi.org/10.7554/eLife.82813.sa2

## Additional files

### Supplementary files
• MDAR checklist

### Data availability
The current manuscript is a computational study, so no data have been generated for this manuscript. Benchmark datasets compiled from the literature are made available as Source Data for figures 2 and 3. Modelling code is publicly accessible through the github repository https://github.com/phbradley/TCRdock, (copy archived at swh:1:rev:060bdb4a59391f2d7d57b0f2a923e4b4d6c9a89f).

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
