## [Editor Report]

The study provides a significant step forward in the prediction of T cell receptor docking to peptide-major histocompatibility complex ligands using a specialised version of the deep neural network structure prediction program AlphaFold. Progress towards this goal has implications for vaccine development, and cancer immunotherapy and is an intrinsically interesting structural problem due to the variability of the T cell receptor scaffold.

---

## [Decision Letter]

**Decision letter after peer review:**

Thank you for submitting your article "Structure-based prediction of T cell receptor:peptide-MHC interactions" for consideration by *eLife*. Your article has been reviewed by 3 peer reviewers, including Michael L Dustin as Reviewing Editor and Reviewer #1, and the evaluation has been overseen by Tadatsugu Taniguchi as the Senior Editor.

Essential revisions:

Please see the comments from each reviewer and the specific recommendations. Please address the points with an emphasis on revisions that will clarify and quantify the strengths of the approach without overstatement and that clarify limitations to identify areas requiring further innovation. Any opportunity to use existing data to check predictions should be prioritised.

*Reviewer #1 (Recommendations for the authors):*

No specific recommendations. In addition to alanine scans, interesting peptide libraries have been developed by changing key residues to all other residues to look for useful altered peptide ligands. Does this version of α fold do well in predicting known APC series- for altered peptide ligand series for OT1 or 1G4 TCR? This is another limited problem where there are currently labor-intensive experimental approaches that might be accelerated by reliable structural predictions.

*Reviewer #2 (Recommendations for the authors):*

Bradley has made a bold step forward here. Yes, TCR-pMHC structure predictions are fraught with danger and have historically been treated with healthy scepticism (and rightly so). However, we need to make a foray into this space – the number of paired TCR sequences is increasing exponentially, but the number of unique experimentally determined TCR-pMHC structures has plateaued. This is an issue for the field. While an argument could be easily made that the experimental data set is insufficient and biased (and it is), I cannot see the experimental data set increasing sufficiently in the next decade – thus someone needs to make a tentative, yet credible step in this direction. The manuscript is clearly lacking any experimental validation of novel predictions – and this represents a key weakness – but the manuscript has merits and will stimulate the field.

1) One major concern is the extent to which biases within the training data would undoubtedly influence predictions – there is clearly an overrepresentation of TCR ternary complexes for HLA-A*02 and H2-Db, for example. How many 'unique' TCR-pMHC complexes out of the limited data set have been determined and what is the functional definition of unique? I feel a graphic highlighting the bias of the current experimental data set in a class I and class II setting would be worthwhile, and discussions surrounding these limitations would be prudent.

2) A key question is whether 'close enough is good enough' regarding structural predictions. In TCR-pMHC structures, subtle shifts in structures (eg < 1A) can be the difference between immune recognition or not. Thus a seemingly close prediction could still miss the mark – this needs reflection.

3) It is also an obvious choice to exclude bulged or longer peptides from the class I modelling in this preliminary work. Typical peptidome data suggests that ~60-80% of HLA-I eluted peptides are 8-11aa long. However, this also means there is an over-representation of shorter epitopes within structural databases (excluding class II). Can the author, therefore, include a comment in the discussion about the extent to which removing the longer peptides might bias the final output towards recapitulating known docking topologies? Would a Docking RMSD vs epitope length be informative? Is modelling more likely to fail as epitopes get larger?

4) There are far more high-resolution pMHC complexes in isolation than co-complexed TCR structures. A complicating factor for modelling of TCR:pMHC complexes will be that any errors modelling epitope structure, particularly in defining which residues project outward will obviously cascade into docking errors. Is it possible to utilise this larger pool of high-resolution structures to better characterise RMSD in peptide modelling?

5) That is can we better understand for which epitopes the modelling pipeline is likely to fail? Bradley has defined cases in which TCR docking topologies diverge. Does the author envisage, or have they tested, similar conditions for which the peptide structure may also fail to be correctly modelled? For example, cases such as PDB: 5T6X where point-mutations in the P3 position drastically alter epitope shape. Or longer epitopes containing helical turns (e.g. 6AVF).

6) Bradley rightly mentions one reversed TCR-pMHC-I docking structure. The original reference is Gras et al. Immunity 2016. Note, there is another reversed structure, TCR-pMHC-II (Beringer et al. NI 2015). Further, as the TCR-pMHC class I docking topologies are very diverse (over 80 degrees – ignoring the reversed docking), there are likely to be many more outliers/poorly modelled ternary complexes beyond the reversed dockers.

The paper is generally well-written and has easy-to-understand figures. The multiple uses of "we" in the abstract comes across oddly with only a single author.

*Reviewer #3 (Recommendations for the authors):*

1. For Figure S1, the gray points do not seem to be specifically described in the figure legend or key. The author should specify what the gray points represent, to avoid ambiguity.

2. Results: "predicted models had displace peptides" should be "predicted models had displaced peptides".

3. Figures 2 and 5 show RMSD values, but the axis labels do not have units (presumably Ångstroms) shown. Appropriate units should be added to all axis labels that are reporting RMSD values.

4. The plot in Figure 2D uses the same colors as Figure 2A and 2B to represent different methods than those panels. The author should change the colors in order not to confuse readers (AF_TCR can be green, AF_TCR finetuned can be some other color).

5. In another study by the author focused on peptide-MHC modeling and reporting the AF finetuning method, the authors appeared to use model_1 and model_2 from AlphaFold, which both utilize templates. However, in this study, the author only appears to use model_2. The author should briefly note why this was done and model_1 was not used here.

6. The author states: "There was a strong positive correlation between predicted and observed model accuracy (Figure 2C).". The author should remove (or replace) "strong" there, as the correlation does not seem strong based on the value (r = 0.484) or visual inspection.

7. It is not clear where the results for the 50 non-binding TCRs are given in any of the figures. If these results are not given, they should be added in the revision, or else the note regarding that set should be removed to avoid confusion.

8. In the "Binding specificity prediction" section of the Results, the statement: "It also lets us investigate a scenario in which we are given not one TCR, but a set of TCRs that all recognize the same epitope, and we consider the extent to which this helps to constrain the target epitope." It seems that if TCRs are known for certain to bind the same epitope, that would be due to an experimental assay that already assesses epitope specificity. Based on the sentence that follows the above sentence, it seems that the author means that the TCRs are inferred to bind the same epitope based on sequence similarity or clustering. If this is the case, the above sentence should be modified to clarify this distinction for readers, e.g. "…a set of TCRs that are predicted to recognize the same epitope…".

9. In the Methods, the author notes: "100 additional 'irrelevant' TCR sequences (50 mouse TCRs and 50 human TCRs) were selected at random from naive CD8 T cells in datasets made publicly available by 10X Genomics (https://www.10xgenomics.com/resources/datasets)." As 10X has many datasets available through that set, the author should specify the specific datasets being utilized for this. The "A new way of exploring immunity" 10X dataset (included by the author in the References) seems to be a part of this, but it seems to be human only and would not contain the 50 murine TCRs.

---

## [Author Response]

Reviewer #1 (Recommendations for the authors):No specific recommendations. In addition to alanine scans, interesting peptide libraries have been developed by changing key residues to all other residues to look for useful altered peptide ligands. Does this version of α fold do well in predicting known APC series- for altered peptide ligand series for OT1 or 1G4 TCR? This is another limited problem where there are currently labor-intensive experimental approaches that might be accelerated by reliable structural predictions.

This is an excellent suggestion. I looked at two examples of APL series with recently published binding data and available structures, for the 1G4 and A6 TCRs (Pettmann et al., *eLife* 2021), as well as T cell activation data for alanine mutants of the Flu M1_58_ epitope (Berkhoff et al., JVI 2005). The results for 1G4 and A6 are not great: with all template information from related structures excluded from the modeling (as was done for the calculations in the main text), there is very little correlation between prediction and experiment, and the structures are also not well-modeled (Figure 6, supplement 1A). If we don't filter out related templates, we see improved structural models (lower docking RMSDs) and a modest but still imperfect correlation between predicted and experimental binding (Figure 6, supplement 1B). This highlights the importance of structural modeling accuracy in binding prediction, as well as the variable performance of the network in this challenging setting of a single TCR binding to multiple highly similar peptides. With a single TCR, rather than a family of related TCRs (as in the epitope decoy discrimination benchmark), there is no opportunity to leverage consensus to average out noise. Better results were seen for the M1_58_ epitope scanning data: predicted binding affinity correlated with levels of T cell activation, albeit still imperfectly.

Reviewer #2 (Recommendations for the authors):Bradley has made a bold step forward here. Yes, TCR-pMHC structure predictions are fraught with danger and have historically been treated with healthy scepticism (and rightly so). However, we need to make a foray into this space – the number of paired TCR sequences is increasing exponentially, but the number of unique experimentally determined TCR-pMHC structures has plateaued. This is an issue for the field. While an argument could be easily made that the experimental data set is insufficient and biased (and it is), I cannot see the experimental data set increasing sufficiently in the next decade – thus someone needs to make a tentative, yet credible step in this direction. The manuscript is clearly lacking any experimental validation of novel predictions – and this represents a key weakness – but the manuscript has merits and will stimulate the field.1) One major concern is the extent to which biases within the training data would undoubtedly influence predictions – there is clearly an overrepresentation of TCR ternary complexes for HLA-A*02 and H2-Db, for example. How many 'unique' TCR-pMHC complexes out of the limited data set have been determined and what is the functional definition of unique? I feel a graphic highlighting the bias of the current experimental data set in a class I and class II setting would be worthwhile, and discussions surrounding these limitations would be prudent.

This is a great suggestion. I've added a new figure (Figure 2, supplement 1) showing the composition of the ternary structure database, for two definitions of redundancy and with respect to three features: species of origin, MHC allele, and peptide length. In the first definition of redundancy ("Set 1"), two TCR:pMHC complexes are considered redundant if the peptides are similar (< 3 mismatches) *and* the TCRs are similar (paired TCRdist<=120). This yields a set of 130 complexes. In the second definition of redundancy ("Set 2"), two TCR:pMHC complexes are considered redundant if the peptides are similar *or* the TCRs are similar. This definition yields a set of 64 complexes.

2) A key question is whether 'close enough is good enough' regarding structural predictions. In TCR-pMHC structures, subtle shifts in structures (eg < 1A) can be the difference between immune recognition or not. Thus a seemingly close prediction could still miss the mark – this needs reflection.

I completely agree: larger numbers of systems and more detailed structural metrics will be necessary to fully understand the method's performance in epitope specificity prediction. I do think that we can conclusively say that this approach is an improvement over the state of the art (AlphaFold multimer and TCRpMHCmodels) in TCR:pMHC ternary structure prediction. From that perspective, the application to epitope prediction, albeit imperfect, represents an important benchmark test.

3) It is also an obvious choice to exclude bulged or longer peptides from the class I modelling in this preliminary work. Typical peptidome data suggests that ~60-80% of HLA-I eluted peptides are 8-11aa long. However, this also means there is an over-representation of shorter epitopes within structural databases (excluding class II). Can the author, therefore, include a comment in the discussion about the extent to which removing the longer peptides might bias the final output towards recapitulating known docking topologies? Would a Docking RMSD vs epitope length be informative? Is modelling more likely to fail as epitopes get larger?

This is an important point. I've added an analysis of peptide backbone and CDR RMSDs broken down by peptide length. For the AF_TCR pipeline, it does look as if the longest peptides in the benchmark (13mers) are modeled rather poorly, but the counts are small.

4) There are far more high-resolution pMHC complexes in isolation than co-complexed TCR structures. A complicating factor for modelling of TCR:pMHC complexes will be that any errors modelling epitope structure, particularly in defining which residues project outward will obviously cascade into docking errors. Is it possible to utilise this larger pool of high-resolution structures to better characterise RMSD in peptide modelling?

Currently these pMHC complexes are being used as templates for the peptide:MHC components. It would be interesting to conduct a more thorough analysis of pMHC structure predictions with these new tools. We look at this a little bit in our peptide-MHC "fine-tuning" paper (Motmaen et al., bioRxiv 2022), but not too deeply since the main focus is on peptide-MHC binding predictions. I also believe that a comprehensive look at pMHC structure modeling is outside the scope of the present paper, but it will be an important step for future work.

5) That is can we better understand for which epitopes the modelling pipeline is likely to fail? Bradley has defined cases in which TCR docking topologies diverge. Does the author envisage, or have they tested, similar conditions for which the peptide structure may also fail to be correctly modelled? For example, cases such as PDB: 5T6X where point-mutations in the P3 position drastically alter epitope shape. Or longer epitopes containing helical turns (e.g. 6AVF).

The analysis of peptide backbone RMSD versus CDR RMSD (Figure 2, supplement 2) gives some insight into this, and it does look as if poor peptide structure predictions tend to have worse interface predictions, although this feature is not assigned a significant *P* value in the linear regression model (Figure 2, supplement 6).

6) Bradley rightly mentions one reversed TCR-pMHC-I docking structure. The original reference is Gras et al. Immunity 2016. Note, there is another reversed structure, TCR-pMHC-II (Beringer et al. NI 2015). Further, as the TCR-pMHC class I docking topologies are very diverse (over 80 degrees – ignoring the reversed docking), there are likely to be many more outliers/poorly modelled ternary complexes beyond the reversed dockers.

I completely agree: the reversed orientation structures are striking outliers, but there are many less-extreme outliers that will likely be poorly modeled by the current protocol. This is nicely illustrated in Figure 2, supplement 6, top left panel, which looks at the correlation between a docking geometry "outlier Z-score" and docking accuracy. More generally, I think TCR structure or binding prediction will always be, at best, an 80% proposition: we can hope for good predictions of the well-behaved 60/70/80% of TCRs, but there will always be challenging cases with long CDR3s/long peptides/divergent binding modes. The key will be to develop robust confidence measures so that we can tell when the predictions are reliable.

The paper is generally well-written and has easy-to-understand figures. The multiple uses of "we" in the abstract comes across oddly with only a single author.

I agree. But it felt even stranger to rewrite the whole thing in the first person singular or avoid first person entirely.

Reviewer #3 (Recommendations for the authors):1. For Figure S1, the gray points do not seem to be specifically described in the figure legend or key. The author should specify what the gray points represent, to avoid ambiguity.

This has been fixed.

2. Results: "predicted models had displace peptides" should be "predicted models had displaced peptides".

This has been fixed.

3. Figures 2 and 5 show RMSD values, but the axis labels do not have units (presumably Ångstroms) shown. Appropriate units should be added to all axis labels that are reporting RMSD values.

This has been fixed.

4. The plot in Figure 2D uses the same colors as Figure 2A and 2B to represent different methods than those panels. The author should change the colors in order not to confuse readers (AF_TCR can be green, AF_TCR finetuned can be some other color).

This has been fixed.

5. In another study by the author focused on peptide-MHC modeling and reporting the AF finetuning method, the authors appeared to use model_1 and model_2 from AlphaFold, which both utilize templates. However, in this study, the author only appears to use model_2. The author should briefly note why this was done and model_1 was not used here.

This has been fixed. We're actually using model_2_ptm now also in the pMHC work, and whenever we need the 'PAE' pairwise confidence measures (although model_1_ptm performs very similarly).

6. The author states: "There was a strong positive correlation between predicted and observed model accuracy (Figure 2C).". The author should remove (or replace) "strong" there, as the correlation does not seem strong based on the value (r = 0.484) or visual inspection.

I replaced "strong" with "significant".

7. It is not clear where the results for the 50 non-binding TCRs are given in any of the figures. If these results are not given, they should be added in the revision, or else the note regarding that set should be removed to avoid confusion.

These background TCRs were used to normalize for pMHC-intrinsic effects. I've added a bit of text to the results to clarify this.

8. In the "Binding specificity prediction" section of the Results, the statement: "It also lets us investigate a scenario in which we are given not one TCR, but a set of TCRs that all recognize the same epitope, and we consider the extent to which this helps to constrain the target epitope." It seems that if TCRs are known for certain to bind the same epitope, that would be due to an experimental assay that already assesses epitope specificity. Based on the sentence that follows the above sentence, it seems that the author means that the TCRs are inferred to bind the same epitope based on sequence similarity or clustering. If this is the case, the above sentence should be modified to clarify this distinction for readers, e.g. "…a set of TCRs that are predicted to recognize the same epitope…".

Great point-- I've changed the wording as suggested.

9. In the Methods, the author notes: "100 additional 'irrelevant' TCR sequences (50 mouse TCRs and 50 human TCRs) were selected at random from naive CD8 T cells in datasets made publicly available by 10X Genomics (https://www.10xgenomics.com/resources/datasets)." As 10X has many datasets available through that set, the author should specify the specific datasets being utilized for this. The "A new way of exploring immunity" 10X dataset (included by the author in the References) seems to be a part of this, but it seems to be human only and would not contain the 50 murine TCRs.

This has been fixed.

Again, I really appreciate the detailed comments and corrections from all the reviewers. Thank you!